# Dietary Inflammatory Index and the Risk of Hyperuricemia: A Cross-Sectional Study in Chinese Adult Residents

**DOI:** 10.3390/nu13124504

**Published:** 2021-12-16

**Authors:** Chen Ye, Xiaojie Huang, Ruoyu Wang, Mairepaiti Halimulati, Sumiya Aihemaitijiang, Zhaofeng Zhang

**Affiliations:** Department of Nutrition & Food Hygiene, School of Public Health, Peking University Health Science Center, Haidian District, Beijing 100191, China; 1510306235@pku.edu.cn (C.Y.); 13161518166@163.com (X.H.); 1710108607@bjmu.edu.cn (R.W.); 2011210145@bjmu.edu.cn (M.H.); 1410606101@pku.edu.cn (S.A.)

**Keywords:** dietary inflammatory index, hyperuricemia, diet, risks

## Abstract

Background: Dietary Inflammatory Index (DII) scores have been consistently associated with several chronic diseases. This study explored the correlation between the DII and hyperuricemia in Chinese adult residents. Methods: The study included 7880 participants from the China Health and Nutrition Survey (CHNS), which was taken in in 2009. A 3-day 24 h meal review method was used to collect diet data and to calculate the DII score. Serum uric acid was obtained to determine hyperuricemia levels. Subjects were divided into a hyperuricemia group and a non-hyperuricemia group, according to their serum uric acid level. Multilevel logistic regression models were used to examine the association between DII scores and hyperuricemia. Results: After adjusting for covariates, a higher DII score was determined to be associated with a higher risk of hyperuricemia. Compared to those in the highest DII score group, the lower DII score group had an inverse association with hyperuricemia risk (Q2: 0.83, 95% CI: 0.70–0.99; Q3: 0.72, 95% CI: 0.60–0.86; Q4: 0.73, 95% CI: 0.61–0.88). The intake of energy-adjusted protein, total fat, MUFAs, PUFAs and saturated fatty acid was higher in the hyperuricemia group. Conclusions: A higher DII score is significantly associated with a higher risk of hyperuricemia. Controlling the intake of pro-inflammatory food may be beneficial to reduce the risk of hyperuricemia.

## 1. Introduction

Hyperuricemia is defined as an increased concentration of uric acid in the blood and occurs due to urate overproduction or because of impaired urate excretion through the kidney and gastrointestinal tract [1,2]. It is considered to be a precursor of gout and associated with other complications such as hypertension [3], chronic kidney disease [4,5], diabetes [6], myocardial congestion and stroke [7]. The prevalence of gout and hyperuricemia is increasing globally and has emerged as a major public health concern in recent years. According to previous studies, the prevalence of hyperuricemia is about 11–17% in the Western population [8] and approximately 5.5% to 23.6% in the Chinese population [9].

Although risk factors for hyperuricemia have not been fully determined, recent studies have shown that chronic inflammation is correlated with hyperuricemia [10,11]. Hyperuricemia is involved in the initiation and progression of inflammation through a complex mechanism. Kidney injury, as the main cause of uric acid excretion disorder, is linked to pro-inflammatory cytokines and markers of inflammation [12]. Some nutrients can also alleviate hyperuricemia nephropathy by reducing the level of inflammation [13].

Diet is one of the main modulators of subclinical inflammation and is associated with hyperuricemia [14,15,16]. Previous research suggests a direct association between specific dietary components (or dietary patterns) and inflammation [17,18]. For instance, red and processed meat, high-fat dairy products and refined grains are pro-inflammatory and are associated with increased levels of high-sensitivity C-reactive protein, interleukin-6 and fibrinogen [17,19,20]. Meanwhile, whole grains, fruit and green vegetables are associated with low levels of inflammatory markers [21]. Previous studies also found that anti-inflammation diets and low inflammatory potential diets were associated with low risks of non-commutable diseases such as obesity, type 2 diabetes mellitus and cardiovascular disease [22].

The Dietary Inflammatory Index (DII) was developed in 2009 and updated in 2014 [23,24], and aims to specifically measure dietary inflammation potential. It is composed of 45 food parameters based on the balance of pro and anti-inflammatory properties of its components. A higher DII score represents a pro-inflammatory diet, whereas a lower DII score represents an anti-inflammatory diet. DII scores are standardized to global dietary intakes, allowing their use across different cultures and dietary patterns.

At present, research on the relationship between DII scores and hyperuricemia in developing countries is very limited. With the growing interest in the role of inflammatory modulation in preventing and controlling hyperuricemia, we conducted a cross-sectional study based on the China Health and Nutrition Survey (CHNS) to investigate the association of DII scores with the risk of hyperuricemia in the Chinese population.

## 2. Materials and Methods

### 2.1. Data Collection and Samples

We used data from the China Health and Nutrition Survey (CHNS), an ongoing open cohort of more than 30,000 subjects from 15 provinces and municipal cities, designed to reflect the nutritional and health status of the Chinese population. The first round of the CHNS started in 1989, and a total of 10 surveys have been carried out so far. Biomarker data were collected in the 2009 survey. Details of the CHNS have been described previously [25]. In this study, the data from the 2009 survey were used.

According to the conclusion of previous relevant studies [8,26], our study included nutrition status and general demographic information. Personal dietary data was collected by a 3-day 24 h meal review, and general demographic information was collected by a face to face interview questionnaire. Of the 18,805 participants in 2009, we identified 9549 subjects with serum uric acid index and dietary data. After excluding participants who were pregnant, or for whom there was incomplete information, 7880 adult subjects were finally included in the study (Figure 1).

The CHNS was approved by the Institutional Review Board of the University of North Carolina at Chapel Hill, and the National Institute for Nutrition and Health (NINH, former National Institute of Nutrition and Food Safety) at the Chinese Center for Disease Control and Prevention (CCDC), and all participants gave written informed consent. 

### 2.2. Definition of Hyperuricemia

Subjects with uric acid (UA) ≥ 420 μmol/L (7 mg/dL) for men, or 360 μmol/L (6 mg/dL) for women were determined to have hyperuricemia [27].

### 2.3. Dietary Inflammatory Index

In brief, the DII was developed after a comprehensive audit of peer-reviewed literature, aimed at measuring dietary inflammatory potential. The China Food Composition Tables Standard Edition (CFCTSD) provided general nutrition data for more than 3000 kinds of foods and ingredients in China. We combined the dietary data obtained from the 3-day 24 h meal review method with the CFCTSD to obtain the data used to calculate the DII.

The DII score in this study was calculated by scoring 28 out of 45 food parameters, including 7 pro-inflammatory parameters and 21 anti-inflammatory parameters. The food parameters included energy, protein, carbohydrates, total fat, saturated fatty acid, cholesterol, monounsaturated fatty acids (MUFAs), polyunsaturated fatty acids (PUFAs), fiber, β-Carotene, vitamin A, vitamin C, thiamin, riboflavin, niacin, folic acid, Fe, Mg, Se, Zn, garlic, ginger, onion, green/black tea, isoflavones, pepper, thyme/oregano and alcohol.

We defined four DII groups according to the quartiles of the DII scores (Q1: −0.49 to 2.30, highest group; Q2, −1.29 to −0.40; Q3, −2.09 to−1.30; Q4 > −4.00 to −2.10, lowest group).

### 2.4. Assessment of Other Characteristics

In addition to diet, the CHNS questionnaire also asked questions on lifestyle and demographic characteristics. In this research, demographic information included gender, age, region (rural and urban), marital status (single, married or other) and education level (none, primary school, lower middle school, upper middle school and above). Current smoking status was divided into two groups: current smokers and non-smokers. Body mass index (BMI) was calculated from the subjects’ physically measured height and weight at the survey.

### 2.5. Statistical Analysis

The descriptive sociodemographic analysis was summarized by the subjects’ gender and hyperuricemia group. Mean values and standard deviations were used for continuous and normally distributed variables. Continuous variables were analyzed by Student’s t-tests and categorical variables were analyzed with the chi-square tests. Multilevel logistic regression models were used to estimate the odds ratios and 95% confidence intervals for the associations between DII groups and hyperuricemia. For the main analysis, two models were established: model 1 was adjusted for age and gender, and model 2 was further adjusted for BMI, region, education level, smoking status and marital status.

The data preprocessing and statistical analyses were all completed by R v4.1.0. All tests were two-sided: alpha was set at 0.05 as the level of significance, and *p* values less than 0.05 were considered significant.

## 3. Results

### 3.1. Subjects Characteristics

There were 7880 subjects included in this study. The average age of subjects was 50.7 years (Appendix A). There were 3628 (46.0%) male participants, 4252 (54.0%) female participants, and 6680 participants (84.8%) were married. There were 4667 (59.2%) participants residing in rural areas, and 2426 (30.8%) had a smoking history. 

In this study, we identified 1235 (15.7%) subjects with hyperuricemia. The prevalence of hyperuricemia was 20.5% in males (748/3628) and 11.5% in females (487/4252). The DII score for the sample ranged from −4.0 to 2.3. The distribution of participants’ characteristics by gender and hyperuricemia status is shown in Table 1.

### 3.2. Comparison of Dietary Intakes between Non-Hyperuricemia and Hyperuricemia Subjects

The total energy and energy-adjusted dietary intakes of carbohydrates, proteins, fats, fiber and vitamins are presented in Table 2. Total energy intakes in the non-hyperuricemia group were higher than the hyperuricemia group, while energy-adjusted protein, fat, MUFAS, PUFAs and saturated fat in the non-hyperuricemia group were significantly lower. Energy-adjusted alcohol intake in the non-hyperuricemia group was significantly higher than that in the hyperuricemia group. The only statistically significant difference in vitamins between groups was that of vitamin B2.

### 3.3. Association between DII Score and Hyperuricemia

Figure 2 shows the results of logistic regression modeling for the association between the DII score and hyperuricemia. Compared to those in the first quartile (Q1, highest DII score group), participants in the other three groups had lower odds of hyperuricemia (Q2: OR 0.80, 95% CI 0.68–0.95; Q3: OR 0.69, 95% CI 0.58–0.82; Q4: OR 0.69, 95% CI 0.58–0.82). The results were similar after adjustment for all of the confounders. The complete results can be found in Appendix A.

The results of the correlation between the DII score and the risk of hyperuricemia (stratified by gender) were shown in Table 3. Both men and women with lower DII scores had a lower risk of hyperuricemia. Compared to the highest DII group, all of the female subjects in the other three groups had lower risks of hyperuricemia (Q2: OR 0.63, 95% CI 0.47–0.84; Q3: OR 0.64, 95% CI 0.48–0.84; Q4: OR 0.66, 95% CI 0.50–0.86) after adjusting for all of the confounders. Male subjects from the lowest and the second-lowest group had a lower risk of hyperuricemia (Q3: OR 0.77, 95% CI 0.62–0.97; Q4: OR 0.75, 95% CI 0.58–0.95). Complete results can be found in Appendix A.

## 4. Discussion

In this cross-sectional analysis, we observed that a higher DII score, which is indicative of a pro-inflammatory diet, was associated with a higher risk of hyperuricemia. These findings remained true after adjustments were made for a range of confounding factors. We also found positive associations between higher intakes of proteins, fats and alcohol with uric acid.

Previous studies have examined the relationship between dietary inflammation and hyperuricemia. A Korean study suggested that women in the highest DII score group had a higher risk of hyperuricemia than those in the lowest group [26], which is comparable to the findings of our study. However, we found that this association was true for both male and female subjects. Our study showed that compared to the highest DII score, all of the female subjects in other groups and male subjects in the lowest and second-lowest groups had lower risks of hyperuricemia. This indicates a strong relationship between dietary inflammatory potential and hyperuricemia in the Chinese population. This may be due to the dietary gap between men and women in the two countries. Traditional Korean diets contain more seafood and ferment foods, while the traditional Chinese diet contains more cereals and livestock meat [28]. Furthermore, the risk of hyperuricemia in female subjects with less dietary inflammatory potential is lower than that in male subjects, suggesting that the association between dietary inflammation and hyperuricemia seems to be stronger in female individuals.

We found that there was a difference in diet between the hyperuricemia and the non-hyperuricemia groups in the Chinese population. The hyperuricemia group had a lower average energy intake and higher energy-adjusted protein and fats. This finding is contrary to previous studies, which have suggested that weight-loss diets may lower serum urate [29]. These data must be interpreted with caution, since weight-loss diets not only have lower calorie intake but also have different proportions of foods and macronutrients. Meanwhile, in the DII calculation, there was no differentiation between plant-sourced proteins and animal-sourced proteins. A higher intake of protein and fat may also suggest a higher intake of animal food, which is thought to be associated with hyperuricemia [30]. We also found that alcohol intake in the hyperuricemia group was higher than the non-hyperuricemia group, which was consistent with other studies. In our study, alcohol intake (1.91 g/d) was much lower than the global daily mean intake in the DII data set (13.98 g/d). Alcohol was considered as an anti-inflammatory parameter in DII calculations, but it is important to note that the abuse of alcohol can seriously damage health.

Dietary factors can affect the development of hyperuricemia and the severity of the symptoms, as well as the inflammatory response. Vitamin C and vitamin E reduce serum uric acid levels through their antioxidant activity [31,32]. Flavonoids reduced uric acid levels, suppressed ROS and protected from kidney damage in several animal studies [33,34]. Inflammation may raise blood uric acid through several potential mechanisms, and oxidative stress is considered to play a crucial role in the inflammatory response. The kidneys play a major role in UA homeostasis, as more than 70% of urate excretion is renal [35]. Chronic inflammation is one of the main mechanisms of renal inter-tubule injury and may ultimately affect the excretion of uric acid [36,37]. We found that a diet’s inflammatory potential can influence hyperuricemia risk, yet further studies are still required to confirm the potential mechanisms and to reveal other possible mechanisms underlying the association between diet, inflammation and uric acid. Our results found that there is the possibility of reducing hyperuricemia and gout through dietary patterns. Given that patients with hyperuricemia and gout often prefer non-pharmacological approaches such as dietary management [38], the adoption of changing the patient’s dietary pattern or dietary guidance will be important considerations.

It is noteworthy that some foods might exert both beneficial and detrimental effects on uric acid levels at the same time. Legumes, for instance, are rich in purine and soybean isoflavones [39]. Previous studies on legumes showed inconsistent evidence of the relationship between legumes and hyperuricemia [16,40]. These plant compounds are often considered as anti-inflammatory factors in the calculation of DII scores, resulting in lower outcomes, but purines in these foods may influence the ultimate effect on hyperuricemia. The effect of purines in food on hyperuricemia was not considered in our study, and further research into this subject is needed.

Strengths of this study include the accuracy of dietary intake data and population-based design. The data were obtained from the CHNS, covering 15 provinces or municipalities, which is more representative of the Chinese residents. The CHNS used a 3-day 24 h meal review to collect diet data, and nutrients in foods were calculated by the CFCTSD and subsequently determined and compiled by the CCDC.

There are some limitations of this study. Firstly, this was a cross-sectional study that cannot account for temporality. Besides this, information about self-aware hyperuricemia or gout history was not available through the questionnaire or interview in the CHNS, and subjects diagnosed with gout or hyperuricemia might change their diet pattern. Therefore, the causality of diet and hyperuricemia cannot be established. Secondly, hyperuricemia was defined by a single blood test of the study population in the CHNS, and the diagnosis in the guideline required two determinations of serum uric acid levels at different times. Therefore, the results of serum uric acid may not represent the true situation. Finally, we included only 28 indicators that were used to calculate the DII, because some of the parameters had insufficient information due to objective conditions. This may have narrowed the effective range of the DII score. Although evidence suggested that the predictive ability was not affected when fewer parameters were used to calculate the DII scores [41], the possibility of affecting the relationship between the DII score and hyperuricemia, due to the lack of parameters, cannot be completely ruled out.

## 5. Conclusions

We documented that a higher DII score was associated with a higher risk of hyperuricemia in China. Increasing anti-inflammatory food and reducing pro-inflammatory food or nutrients in daily life might be a potential strategy to reduce uric acid.

## Figures and Tables

**Figure 1 nutrients-13-04504-f001:**
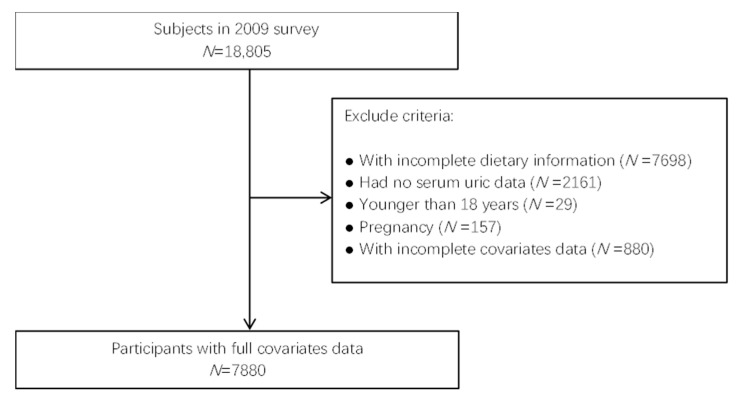
Flow chart for inclusion and exclusion of research subjects.

**Figure 2 nutrients-13-04504-f002:**
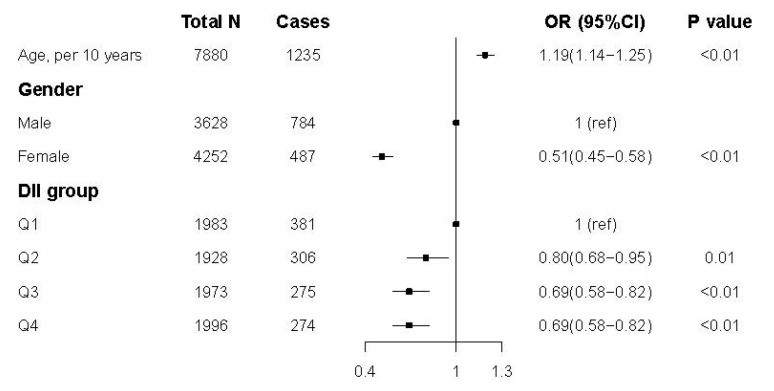
Odds ratios (ORs) and 95% confidence intervals for hyperuricemia in CHNS.

**Table 1 nutrients-13-04504-t001:** Characteristics of subjects according to gender and hyperuricemia status.

Characteristics	Male				Female			
	Total (*n* = 3628)	Non-Hyperuricemia (*n* = 2880)	Hyperuricemia (*n* = 748)	*p* Value	Total (*n* = 4252)	Non-Hyperuricemia (*n* = 3765)	Hyperuricemia	*p* Value
Age (years)	50.8(15.0)	51.0 (15.0)	50.3 (15.1)	0.29	50.6(14.8)	49.6 (14.6)	58.2 (14.0)	
DII ^1^ group 1				<0.01				<0.01
Q1 (−0.49 to 2.30)	1081	828 (77%)	253 (23%)		902	774 (86%)	128 (14%)	0.01
Q2 (−1.29 to −0.40)	926	718 (78%)	208 (22%)		1002	904 (90%)	98 (10%)	
Q3 (−2.09 to −1.30)	886	724 (82%)	162 (18%)		1087	974 (90%)	113 (10%)	
Q4 (−4.00 to −2.10)	735	609 (83%)	126 (17%)		1261	1113 (88%)	148 (12%)	
Marital status				0.89				<0.01
Single	279	224 (80%)	55 (20%)		147	136 (93%)	11 (7%)	
Married	3129	2482 (79%)	647 (21%)		3551	3175 (89%)	376 (11%)	
Other	220	173 (79%)	47 (21%)		554	454 (82%)	100 (18%)	
Region				<0.01				<0.01
Urban	1516	1138 (75%)	378 (25%)		1697	1453 (86%)	244 (14%)	
Rural	2112	1741 (82%)	371 (18%)		2555	2312 (90%)	243 (10%)	
Education				<0.01				<0.01
None	528	429 (81%)	99 (19%)		1329	1133 (85%)	196 (15%)	
Primary school	703	568 (81%)	135 (19%)		865	772 (89%)	93 (11%)	
Lower middle school	1375	1117 (81%)	258 (19%)		1215	1099 (90%)	116 (10%)	
Upper middle school and above	1022	765 (75%)	257 (25%)		843	761 (90%)	82 (10%)	
Current smoke status				0.17				0.63
No	1381	1079 (78%)	302 (22%)		4073	3609 (89%)	464 (11%)	
Yes	2247	1800 (80%)	447 (20%)		179	156 (87%)	23 (13%)	
BMI ^2^ (kg/m^2^)				<0.01				<0.01
≤18.5	1935	198 (10%)	18 (1%)		2284	245 (11%)	16 (1%)	
18.5–24	216	1636 (757%)	299 (138%)		261	2093 (802%)	191 (73%)	
24–28	1157	833 (72%)	324 (28%)		1254	1084 (86%)	170 (14%)	
≥28	320	212 (66%)	108 (34%)		453	343 (76%)	110 (24%)	

^1^ DII: dietary inflammatory index; ^2^ BMI: body mass index.

**Table 2 nutrients-13-04504-t002:** Dietary intakes of energy and energy-adjusted dietary intakes (per 1000 kcal).

Nutrients	Non-Hyperuricemia	Hyperuricemia	*p*
Energy (kcal)	1919.38 ± 570.52	1879.58 ± 590.00	0.03
Carbohydrate (g/1000 kcal)	159.51 ± 54.98	153.51 ± 54.48	0.11
Protein (g/1000 kcal)	38.93 ± 1.49	42.44 ± 1.57	<0.01
Fat (g/1000 kcal)	21.47 ± 1.30	24.41 ± 1.37	<0.01
Cholesterol (mg/1000 kcal)	300.77 ± 28.8	302.04 ± 28.5	0.39
MUFAs ^1^ (g/1000 kcal)	3.72 ± 0.32	4.10 ± 0.33	<0.01
PUFAs ^2^ (g/1000 kcal)	4.09 ± 0.34	4.23 ± 0.33	0.02
Saturated fatty acid (g/1000 kcal)	10.73 ± 2.88	11.44 ± 2.84	<0.01
Fiber (g/1000 kcal)	10.86 ± 0.99	16.64 ± 1.45	0.16
Vitamin A (RAE/1000 kcal)	453.41 ± 77.17	484.82 ± 80.47	0.10
Vitamin B1 (mg/1000 kcal)	0.52 ± 0.21	0.52 ± 0.20	0.10
Vitamin B2 (mg/1000 kcal)	0.47 ± 0.21	0.48 ± 0.21	<0.01
Vitamin C (mg/1000 kcal)	48.05 ± 3.05	47.97 ± 2.68	0.28
Alcohol (g/1000 kcal)	0.92 ± 0.70	1.52 ± 0.56	<0.01

^1^ MUFAs: monounsaturated fatty acids; ^2^ PUFAs: polyunsaturated fatty acids.

**Table 3 nutrients-13-04504-t003:** Odds ratios (ORs) and 95% confidence intervals for hyperuricemia and DII scores stratified by gender.

DII ^1^ Group	Male	Female
	OR (95% CI)	OR (95% CI)
Q1	1 (ref)	1(ref)
Q2	1.00 (0.81, 1.24)	0.63 (0.47, 0.84)
Q3	0.77 (0.62, 0.97)	0.64 (0.48, 0.84)
Q4	0.75 (0.58, 0.95)	0.66 (0.50, 0.86)

^1^ DII: Dietary Inflammatory Index.

## Data Availability

Publicly available datasets were analyzed in this study. This data can be found here: https://www.cpc.unc.edu/projects/china/data (accessed on 11 December 2021).

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
