# Peer review of "Dietary Inflammatory Index and the Risk of Hyperuricemia: A Cross-Sectional Study in Chinese Adult Residents"

_nutrients, 2021, doi:10.3390/nu13124504_

Round 1

Reviewer 1 Report

I thank the authors for their time and efforts in preparing this manuscript. The authors present a cross-sectional analysis of the dietary inflammatory index scores and prevalence of hyperuricemia in Chinese study participants, with an emphasis on energy content, macronutrients and some minor nutrients. This study provides a good basis for further study into the role of DII on hyperuricemia and is an important work for promoting the study of dietary contributions to health and disease. The use of statistics is appropriate and balanced. The strengths and limitations of the current study are clearly described. Study may benefit from further details of the specific MUFAs and PUFAs present in participant diets, as these may vary widely in their DII contributions. I offer the following suggestions to the authors to enhance the manuscript in its current form.

Table 3 does not clearly show results and should be reorganized for clarity. Missing data should be explained.

The authors must be very careful while offering suggestions, as observed in line 191. While these results show associations between DII and hyperuricemia, an intervention is necessary to claim that reducing DII may also provide benefits in individuals with hyperuricemia. Please reword.

Writing is very good, but it is recommended to double check language and syntax throughout for clarity and accuracy. Below are some examples of suggested edits.

Line 42. Changes “researches” to “research” to correct for plural form.

Sentence starting on Line 43: revise. It appears that it should say “…are pro-inflammatory and CONTAIN high levels of…”

Line 48 and 49: T2DM and CVD need to be spelled out in their first appearance in the text. Please double check abbreviations and acronyms throughout.

Line 130: this should say “Total energy intake in non-hyperuricemia group…”

Author Response

Point 1: Table 3 does not clearly show results and should be reorganized for clarity. Missing data should be explained.

Response 1: Sorry for my mistake. We updated Table 3. Confounding factors that not included in model 1 were blank in Table 3.

(Figure 2, Table S2)

Point 2: The authors must be very careful while offering suggestions, as observed in line 191. While these results show associations between DII and hyperuricemia, an intervention is necessary to claim that reducing DII may also provide benefits in individuals with hyperuricemia. Please reword.

 Response 2: Thank you for your suggestion. We reworded this part.

(Line 177-178)

Point 3: Line 42. Changes “researches” to “research”; Sentence starting on Line 43: revise; Line 48 and 49: T2DM and CVD need to be spelled out in their first appearance in the text; Line 130: this should say “Total energy intake in non-hyperuricemia group…”

 Response 3: Sorry for my mistake. We corrected grammar and spelling mistakes.

(Line 34, 43-44, 48, 137)

Reviewer 2 Report

The authors have conducted a cross-sectional study looking into the association between inflammatory potential of diet and hyperuricemia. Please re-review the language and style carefully in the manuscript and make necessary changes or use an editing service for the same.

I have the following comments:

  1. Line 20: remove "I" at the beginning of conclusions.
  2. Line 22: replace "would be more" with "may" and "controlling" with "reduce".
  3. Line 29: Replace "cause" with "associated with".
  4. Obesity was studies as a modifier in the reference provided not casually related to hyperuricemia. Following the correction in Line 29 would address this.
  5. Line 36: Remove "ascertained".
  6. Line 45: expand hsCRP.
  7. Line 48 and 49: expand T2DM and CVD.
  8. Line 53-54: removed the word "more".
  9. Line 58-60: Restate the sentences and review language and style.
  10. Line 64: First sentence is redundant.
  11. Line 65: State when did the CHNS start.. the authors only mentioned that it is an ongoing study.
  12. Line 74: replace apostrophe with comma in the participant total.
  13. Line 80-81: Use the abbreviations that were included earlier. Alternately, remove the mention of the abbreviations in the earlier paragraph.
  14. Section 2.2: Review and rewrite this section avoiding redundancy.
  15. Line 98-99: Expand and provide abbreviations for the components of DII.
  16. Section 2.4: Did the authors directly include the variables from previous literature as covariates without testing for potential confounding? Why were physical activity and total energy intake not considered?
  17. Line 103: Expand BMI.
  18. Line 107: replace quintiles with quartiles.
  19. The authors mentioned about continuous variables but did not say anything about categorical variables.
  20. Similarly how did the authors analyze non-normally distributed variables.
  21. Line 111: A model including age and gender is not a univariate model
  22. Line 115: Mention that alpha was set at 0.05 instead of p-value for statistical significance
  23. Line 118: Why is median age mentioned instead of mean and standard deviation?
  24. Line 120: replace "with" with "had".
  25. Line 122-123: Restate the sentence about prevalence among males and females.
  26. Table 1 does  not show any p-values. There is no associated summary in the results section 3.1
  27. Do the categories of BMI correspond to normal, overweight and obese categories in Chinese population?
  28. Table 2: Replace "Smoke" with "Smoking status"  - clarify whether these categories are "Ever" and "Never" smoking or current smoking (yes/no) at baseline. Also in Table 1.
  29. Table 2: Replace "Drink" with "Alcohol drinking" There are no results provided here. What does "0.77" under p-value represent?
  30. If Model 2 results are not different than Model 1, then there is no need to present those results in the main tables. They can be presented as a Supplementary table
  31. The authors present the stratifies results in Table 2 but do not mention any summary in the Results section
  32. Why did the authors not consider the lowest DII category as the reference? That would show a positive association between hyperuricemia and DII, which would be more sensible.
  33. Line 153: remove "more"
  34. Line 159: Restate this sentence
  35. Line 161: replace "in across all genders" with "among men and women".
  36. Line 168-170: The authors can't make this conclusions from the results in Table 4. Based on the previously stated results it can be stated that there seems to be a stronger association between diet and hyperuricemia among women.
  37. Line 180-181: Provide references to these statements
  38. Line 192-193: Provide reference to this statement

Author Response

Point 1: Line 20: remove "I" at the beginning of conclusions; Line 22: replace "would be more" with "may" and "controlling" with "reduce"; Line 29: Replace "cause" with "associated with"; Line 36: Remove "ascertained"; Line 53-54: removed the word "more"; Line 74: replace apostrophe with comma in the participant total; Line 80-81: Use the abbreviations that were included earlier. Alternately, remove the mention of the abbreviations in the earlier paragraph; Line 107: replace quintiles with quartiles; Line 120: replace "with" with "had"; Line 122-123: Restate the sentence about prevalence among males and females; Table 2: Replace "Smoke" with "Smoking status"  - clarify whether these categories are "Ever" and "Never" smoking or current smoking (yes/no) at baseline. Also in Table 1; Line 153: remove "more"; Line 159: Restate this sentence; Line 161: replace "in across all genders" with "among men and women".

 Response 1: Sorry for my mistake. We corrected grammar and spelling mistakes.

(Line 20, 22, 28, 35, 52, 71, 77-78, 98, 124, 126-127, Table 1, Table S1, Table S2, 162, 167-168, 169)

Point 2: Obesity was studies as a modifier in the reference provided not casually related to hyperuricemia. Following the correction in Line 29 would address this

Response 2: Sorry for my mistake, we deleted this modifier.

(Line 30)

Point 3: Line 64: First sentence is redundant; Line 65: State when did the CHNS start.. the authors only mentioned that it is an ongoing study.

Response 3: Thanks for your suggestion. We rewrote this paragraph to state the CHNS better.

(Line 62-65)

Point 4: Review and rewrite this section avoiding redundancy.

Response 3: Thanks for your suggestion. We rewrote this paragraph.

(Line 81-83)

Point 5: Line 45: expand hsCRP; Line 48 and 49: expand T2DM and CVD; Line 98-99: Expand and provide abbreviations for the components of DII; Line 103: Expand BMI;

Response 3: We checked and corrected the abbreviations

(Line 44, 48, 95, 106)

Point 6: Did the authors directly include the variables from previous literature as covariates without testing for potential confounding? Why were physical activity and total energy intake not considered

Response 6: The selection of potential variables referred to previous studies and we have tested potential confounding in our previous study. Total energy intake was concluded in the calculation of DII. As for physical activity, among 1,1929 subjects in the CHNS 2009 lifestyle subset, only 1302 subjects reported their physical activities time and 5663 subjects reported their working physical load (light physical activities job, moderate physical activities job and heavy physical activities job). Due to the small number size and the negative results in some of previous studies, we decided to delete this covariate.

Point 7: The authors mentioned about continuous variables but did not say anything about categorical variables; Similarly how did the authors analyze non-normally distributed variables; Mention that alpha was set at 0.05 instead of p-value for statistical significance

Response 7: I apologize for my unclear express. We rewrote statistical analysis part according to your suggestion.

(Line 111)

Point 8: Why is median age mentioned instead of mean and standard deviation

Response 8: Sorry for the misuse of words. Age was presented by mean and standard deviation.

Point 9: Table 1 does not show any p-values. There is no associated summary in the results section 3.1

Response 9: We redrawn Table 1 according to the suggestion from other reviewers to present the data better. P value is shown in Table 1 right now.

(Table 1)

Point 10: Do the categories of BMI correspond to normal, overweight and obese categories in Chinese population?

Response 10: Yes, the categories of BMI correspond to underweight, normal, overweight and obese categories in Chinese population according to Chinese guide.

Point 11: If Model 2 results are not different than Model 1, then there is no need to present those results in the main tables. They can be presented as a Supplementary table

Response 10: Thank you for your suggestion. We adjusted the presentation of the data.

(Figure 2, Table S1)

Point 12: Why did the authors not consider the lowest DII category as the reference? That would show a positive association between hyperuricemia and DII, which would be more sensible

Response 12: We wished to see the protective effect due to reduce dietary inflammatory potential. Besides, using the lowest DII group as the reference didn’t as sensible as using the highest group.

Point 13: Line 168-170: The authors can't make this conclusions from the results in Table 4. Based on the previously stated results it can be stated that there seems to be a stronger association between diet and hyperuricemia among women.

Response 13: Thanks to your suggestion. We reworded this sentence.

(Line 177-178)

Point 14: Line 180-181: Provide references to these statements

Response 14: Sorry for my negligence. We added references to these statements

(line 188)

Point 15: Line 192-193: Provide reference to this statement

Response 15: The sentence in line 192-193 is a statement about our results. But we think you might mean the sentence below it. We added reference to that statement

(line 208)

Reviewer 3 Report

The paper “Dietary Inflammatory Index and the Risk of hyperuricemia: A 2 Cross-sectional Study in Chinese Adult Residents” investigates the association between dietary inflammatory pattern and prevalence of hyperuricemia. The paper is simple and clearly written. Given the cross-sectional nature of the study, unfortunately, not much can be drawn about causality of DII in the development of hyperuricemia, and either about reverse causality. Nevertheless, the study is indeed interesting. A few suggestions to improve its soundness:

  • Intro:
    • Line 45: “… high levels of hsCRP, interleukin-6 and fibrinogen”. This sentence doesn’t sound correct. Please, rephrase
  • Material and Methods:
    • Line 74, typo: “18’805”
    • Line 84-85: “…consensus of multidisciplinary experts on diagnosis and treatment of hyperuricemia related diseases in China”. This sentence doesn’t sound correct. Please, rephrase
    • Figure 1 – flow chart. Please, include the exclusion criteria in the boxes
    • Is DII adjusted for Energy?
    • Line 102: needs reference(s) for “…previous relevant studies”
    • Line 113: why adjusting for “marital status”? Evidence of being a cofounder
  • Results
    • Line 122-123: “Prevalence was higher…”. Also, how would you explain that the hyperuricemia was higher in males, even though 29% (1081/3628 [males in Q1/males]) of them (the biggest %) is in the Q1 of DII? You could plot a 2x2 table of DII/gender to have a clearer distribution of DII across sexes
    • Line 130: “…hyperuricemia…” should be “non- hyperuricemia”
    • Table 2: how do you explain that the “Hyperuricemia” group has lower Energy intake but higher intake in grams of basically all food macronutrients? Moreover, alcohol should be included since it is used in the DII calculation, and it could explain the difference in energy intake
    • Line 140-143: the results listed are from adjusted model, while I think the authors were meant to list the unadjusted model results
    • Is there a typo for the drinker category in table 3?
    • Table 4: is the model adjusted? If no, results for the adjusted should be reported
  • Discussion:
    • Line 192-193: The sentence “Given that patients with hyperuricemia and gout often prefer non-pharmacological approaches…” needs a reference
    • Line 198: “previous” in capital letter
    • Line 206: I would actually see the “3-day 24-h meal” as a weakness. Three days do not always sufficiently proxy for a dietary pattern.
    • Line 209: While the authors are not implying any causality of DII, it should also be mentioned that given the cross-sectional nature of the study, it can’t be ruled out that a diagnosis of hyperuricemia might have led to a change in the population dietary pattern
    • Line 217: could the reduced list of food might have instead overestimated the association? To back up your hypothesis of “underestimation”, could you please add more info on your dietary index by including how many anti- and how many pro- inflammatory items were included in your study?

Hope you find the above comments useful. Best of luck!

Author Response

Point 1: Line 45, Line 74, Line 84-85: This sentence doesn’t sound correct; Line 130: “…hyperuricemia…” should be “non- hyperuricemia”; Line 198: “previous” in capital letter; Table 3: is there a typo for the drinker category in table 3?

 Response 1: Sorry for my mistake. We corrected grammar and spelling mistakes.

(line 43-44, 71, 134, 212, Table S3)

Point 2: Figure 1 – flow chart. Please, include the exclusion criteria in the boxes

 Response 2: Thanks for your suggestion. We added exclude criteria in the chart.

(figure 1)

Point 3: Is DII adjusted for Energy

 Response 3: No, we didn’t adjusted DII for energy, but energy is included in the calculation of DII (Overall inflammatory effect score: 0.180)

Point 4: Needs reference(s) for Covariates and why adjusting for “marital status”

 Response 4: Sorry for my negligence. We added references and move it to the front paragraph. As for “marital status”, we included it because some previous studies contained marital status and we found that marital status was a confounding factor in Chinese population.

(line 68)

Point 5: hyperuricemia was higher in males

 Response 5: Thank you for your suggestion. We changed table 1 into a 2*2 table of hyperuricemia/gender. Lots of previous studies in different countries suggested that men are at higher risk of gout or hyperuricemia [1]. This might because of the uricosuric effects of oestrogen which partly explained why prevalence of gout or hyperuricemia in female population usually increased from the late-menopausal transition stage. Our results were consisted with this explanation. we found out that the average age of female hyperuricemia group (58.2 years) was significantly higher than male hyperuricemia group (51.0) and female non-hyperuricemia group (49.6).

(Table 1)

[1] Dehlin, M., Jacobsson, L., & Roddy, E. (2020). Global epidemiology of gout: prevalence, incidence, treatment patterns and risk factors. Nature reviews. Rheumatology, 16(7), 380–390. https://doi.org/10.1038/s41584-020-0441-1

Point 6: Table 2: how do you explain that the “Hyperuricemia” group has lower Energy intake but higher intake in grams of basically all food macronutrients? Moreover, alcohol should be included since it is used in the DII calculation, and it could explain the difference in energy intake

 Response 6: Thanks for your meticulous reminder. We accidentally uploaded incomplete and unverified version. We updated Table 2 and added alcohol intake according to your suggestion. Energy-adjusted carbohydrate was lower in hyperuricemia group, which could partly explain the difference in the energy ratio of macronutrients, but had no statistical significance. We also added alcohol intakes in discussion section.

(Table 2)

Point 7: Line 140-143: the results listed are from adjusted model, while I think the authors were meant to list the unadjusted model results

 Response 7: Sorry for my mistake. We revised the sentence.

(line 144-145)

Point 8: Table 4: is the model adjusted? If no, results for the adjusted should be reported

 Response 8: I’m so sorry because I didn’t make it clear. The models used in stratified analysis has been adjusted. We added explanations in line 155

Point 9: Line 192-193: The sentence “Given that patients with hyperuricemia and gout often prefer non-pharmacological approaches…” needs a reference

 Response 9: According to your suggestion, we added the reference to this sentence.

(line 208)

Point 10: I would actually see the “3-day 24-h meal” as a weakness. Three days do not always sufficiently proxy for a dietary pattern

Response 10: We agree that the “3-day 24 meal” can’t fully represent a dietary pattern. But in this article this might not be the weakness. Firstly, DII calculations require accurate food intakes. Therefore, in practice, DII can only be used in 24h dietary recall interview-derived or food record data. In CHNS, household food consumption was determined by examining changes in inventory from the beginning to the end of each day, in combination with a weighing and measuring technique. All purchases, home production, and processed snack foods have been recorded. This ensured the accuracy of the dietary data.

Point 11: While the authors are not implying any causality of DII, it should also be mentioned that given the cross-sectional nature of the study, it can’t be ruled out that a diagnosis of hyperuricemia might have led to a change in the population dietary pattern

Response 11: Thanks to your suggestion. We added this possibility of bias into the limit part.

(line 226-226)

Point 12: could the reduced list of food might have instead overestimated the association? To back up your hypothesis of “underestimation”, could you please add more info on your dietary index by including how many anti- and how many pro- inflammatory items were included in your study

Response 12: Thanks for your suggestion. We added the number of items we included in the methods part. As for association, although previous evidence suggested that the predictive ability was not affected when less than 30 dietary parameters were used to calculate DII scores, our score should be higher than real result for most parameters that not available in our study are anti-inflammatory. The impact caused by the reduced of parameters was uncertain so we rewrite the sentence to avoid ambiguity.

(line 93, 234)

Round 2

Reviewer 2 Report

Thank you for addressing previous comments and explanation about controlling for energy intake and physical activity. As has been suggested previously, the authors should read through the manuscript before submission for spelling mistakes and language.

I have the following comments:

  1. Line 63: The hyphen should be removed in the word de-signed
  2. Line 65: The hyphen should be removed in the word de-tails
  3. Line 67: Please reword the sentence to avoid redundancy "... because information on biomarkers was only available in 2009"
  4. Line 81: please delete the sentence in this line. The second sentence is good enough with the reference [27] at the end of the sentence
  5. Line 102: please remove the hyphen in the word "gen-der"
  6. Table 1: please replace the heading for the column of p-values among females.
  7. Table 1: Please change to "Current smoking status"
  8. Table 1: Please include expansions for DII in the foot notes of the table unless it may have cut off in the PDF format.
  9. Table 2: Please provide expansions for MUFA and PUFA in the foot notes of Table 2
  10. Figure 2 might need a better resolution for publication
  11. Line 145: Please insert space between 95% and CI
  12. Table 3: There is no need to provide both confidence intervals and p-values. Please insert spaces uniformly between the confidence interval values and between OR and beginning of parentheses
  13. Line 223: "... this study was a cross-sectional study..."
  14. Line 235: Did the authors intend to add some thing in the parentheses in a local language or was it an oversight?

Author Response

Point 1: Line 63: The hyphen should be removed in the word de-signed; Line 65: The hyphen should be removed in the word de-tails; Line 67: Please reword the sentence to avoid redundancy "... because information on biomarkers was only available in 2009"; Line 81: please delete the sentence in this line. The second sentence is good enough with the reference [27] at the end of the sentence;  Line 102: please remove the hyphen in the word "gen-der"; Table 1: please replace the heading for the column of p-values among females; Table 1: Please change to "Current smoking status"; Line 145: Please insert space between 95% and CI; Line 223: "... this study was a cross-sectional study..."

Response 1: Sorry for my mistake. We corrected grammar mistakes and reworded unclear sentences.

(Line 63, 65-67, 66, 81, 103, Table 1, 148, 227)

Point 2: Table 3: There is no need to provide both confidence intervals and p-values. Please insert spaces uniformly between the confidence interval values and between OR and beginning of parentheses

Response 2: Thanks for your suggestion. We changed Table 3.

(Table 3)

Point 3: Table 1: Please include expansions for DII in the foot notes of the table unless it may have cut off in the PDF format; Table 2: Please provide expansions for MUFA and PUFA in the foot notes of Table 2

Response 3: Thanks for your suggestion. We expanded the abbreviations.

(Table 1, Table 2)

Point 4: Figure 2 might need a better resolution for publication

Response 4: According to your suggestion, we improved the resolution of the picture. We can also submit PDF images to avoid low resolution.

(Figure 2)

Point 5: Line 235: Did the authors intend to add some thing in the parentheses in a local language or was it an oversight?

Response 5: Thanks for your reminder. It was an oversight, and we added the right reference here.

(Line 238)